# Infrastructure for AI Agents

**Alan Chan**[*]
*Centre for the Governance of AI*

**Kevin Wei**
*Harvard Law School*

**Sihao Huang**
*University of Oxford*

**Nitarshan Rajkumar**
*University of Cambridge*

**Elija Perrier**
*Australian National University*

**Seth Lazar**[†]
*Australian National University*

**Gillian K. Hadfield**[†]
*Johns Hopkins University*

**Markus Anderljung**[†]
*Centre for the Governance of AI*

**Reviewed on OpenReview:** *https://openreview.net/forum?id=Ckh17xN2R2*

## Abstract

**AI agents** plan and execute interactions in open-ended environments. For example, OpenAI's Operator can use a web browser to do product comparisons and buy online goods. To facilitate beneficial interactions and mitigate harmful ones, much research focuses on directly modifying agent behaviour. For example, developers can train agents to follow user instructions. This focus on direct modifications is useful, but insufficient. We will also need external protocols and systems that shape how agents interact with institutions and other actors. For instance, agents will need more efficient protocols to communicate with each other and form agreements. In addition, attributing an agent's actions to a particular human or other legal entity can help to establish trust, and also disincentivize misuse. Given this motivation, we propose the concept of **agent infrastructure**: technical systems and shared protocols external to agents that are designed to mediate and influence their interactions with and impacts on their environments. Just as the Internet relies on protocols like HTTPS, our work argues that agent infrastructure will be similarly indispensable to ecosystems of agents. We identify three functions for agent infrastructure: 1) attributing actions, properties, and other information to specific agents, their users, or other actors; 2) shaping agents' interactions; and 3) detecting and remedying harmful actions from agents. We provide an incomplete catalog of research directions for such functions. For each direction, we include analysis of use cases, infrastructure adoption, relationships to existing (internet) infrastructure, limitations, and open questions. Making progress on agent infrastructure can prepare society for the adoption of more advanced agents.

---

[*]Correspondence to alan.chan@governance.ai
[†]Senior authors.

# 1 Introduction

A fundamental goal of the AI research community is to build **AI agents**: AI systems that can plan and execute interactions in open-ended environments,[1] such as making phone calls or buying online goods (Maes, 1994; 1995; Lieberman, 1997; Jennings et al., 1998; Johnson, 2011; Sutton & Barto, 2018; Russell & Norvig, 2021; Chan et al., 2023; Shavit et al., 2023; Wu et al., 2023; OpenAI, 2018; Gabriel et al., 2024; Kolt, 2024; Lazar, 2024). Agents differ from other computational systems in two significant ways. First, in comparison to foundation models used as chatbots, agents directly interact with the world (e.g., a flight booking website) rather than only with users. Second, in comparison to traditional software (e.g., an implementation of a sorting algorithm), agents can adapt to under-specified task instructions. Although the AI community has been developing agents for decades, these agents typically performed only a narrow set of tasks (Wooldridge, 2009; Mnih et al., 2013; Silver et al., 2018; Badia et al., 2020). In contrast, recent agents built upon language models can attempt—with varying degrees of reliability (Kapoor et al., 2024; Liu et al., 2023; Mialon et al., 2023; Lu et al., 2024; Zhang et al., 2024)—a much wider array of tasks, such as software engineering (Jimenez et al., 2024; Wu, 2024; Chowdhury et al., 2024) or office support (Gur et al., 2024; MultiOn, 2024).

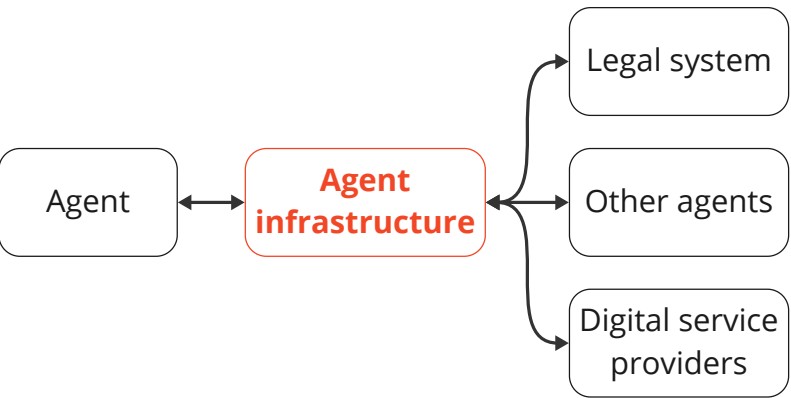

Figure 1: **Agent infrastructure** consists of technical systems and shared protocols external to agents that are designed to mediate and influence their interactions with and impacts on their environments, including interactions with existing institutions (e.g., legal and economic systems) and actors (e.g., digital service providers, humans, other AI agents).

More general-purpose agents could automate a wide range of both beneficial and harmful tasks. Personalized agents could help individuals make a wide variety of difficult decisions, such as what insurance to buy or which school to attend (Van Loo, 2019; Sunstein, 2024; Lazar et al., 2024). Deployment of agents throughout the economy could lead to productivity growth (Korinek & Suh, 2024). Yet, barriers such as a lack of reliability, an inability to maintain effective oversight, or the absence of recourse mechanisms could hinder beneficial adoption. On the other hand, such barriers may not be critical for motivated malicious actors. Potential issues include scams (Fang et al., 2024b; Chen & Magramo, 2024) and large-scale disruption of digital services (Fang et al., 2024a; Bhatt et al., 2023; U.S. Attorney's Office, 2024).

To facilitate beneficial tasks and mitigate harmful ones, much AI research focuses on **system-level** interventions, which intervene on the AI system itself to shape its behaviour. Major areas of work include goal specification and following (Hadfield-Menell et al., 2016; Christiano et al., 2017; Leike et al., 2018; Bai et al., 2022; Hua et al., 2024; Wang et al., 2024a;b; Kirk et al., 2024; Huang et al., 2024; OpenAI, 2024), adversarial robustness (Greshake et al., 2023; Tamirisa et al., 2024; Zou et al., 2024; Anil et al., 2024; Wallace et al., 2024), and cooperation (Lerer & Peysakhovich, 2019; Hu et al., 2020; Leibo et al., 2021; Dafoe et al., 2021). System-level interventions, if adopted (Askell et al., 2019), can help to improve the reliability of agents, but they may not be sufficient for facilitating beneficial adoption or mitigating risks. For example, the difficulty

---

[1]Gabriel et al. (2024) heavily inspire this definition. See (Maes, 1994; Sutton & Barto, 2018; Chan et al., 2023; Shavit et al., 2023; Kolt, 2024; Lazar, 2024) for other definitions.

| Function | Research directions |
|---|---|
| **Attribution** (Section 3): Attributing actions, properties, and other information to agents or users | • **Identity binding** (Section 3.1): Associates an agent or its actions with a legal identity, such as a human or corporation.

• **Certification** (Section 3.2): Makes, verifies, and revokes claims about an agent (instance), such as what data the agent is collecting, which tools it can access, and the level of autonomy it has been authorized to exercise.

• **Agent IDs** (Section 3.3): Identifies instances of agents and links to useful attributes, such as the credentials above. |
| **Interaction** (Section 4): Shaping how agents interact with counterparties. | • **Agent channels** (Section 4.1): Isolates agent traffic from all other digital traffic in interactions with an existing digital service (e.g., Airbnb).

• **Oversight layers** (Section 4.2): Enables actors (e.g., a user) to intervene upon an agent's actions.

• **Inter-agent communication** (Section 4.3): Helps facilitate joint activities amongst groups of agents.

• **Commitment devices** (Section 4.4): Enforce commitments between agents. |
| **Response** (Section 5): Addressing problems that occur during interaction with an agent. | • **Incident reporting** (Section 5.1): Enables actors (e.g., humans, agents interacting with other agents) to report incidents.

• **Rollbacks** (Section 5.2): Helps void or undo an agent's actions. |

Table 1: This table summarizes all of the types of agent infrastructure that we discuss in this work.

of achieving adversarial robustness could mean that companies need additional assurances before adopting agents to complete economically valuable tasks. In particular, potential assurance schemes include certification of agents, insurance, or identity solutions that establish trust between different parties using agents. Such tools shape how agents interact with institutions (e.g., legal and economic systems) and other actors (e.g., web service providers, humans, other AI agents).

Given the insufficiency of system-level interventions, we propose the concept of **agent infrastructure**: technical systems and shared protocols external to agents[2] that are designed to mediate and influence their interactions with and impacts on their environments.[3] These systems and protocols could be novel, or simply extensions of existing systems and protocols. Examples of agent infrastructure include inter-agent communication protocols (Marro et al., 2024), IDs for agents (Chan et al., 2024b), systems for certifying an agent's properties or behaviour, and methods for rolling back an agent's actions (Patil et al., 2024). We include more examples in Table 1. Our notion of agent infrastructure is *not* about the technical systems that enable basic operation of agents (e.g., memory systems, cloud compute), although it will in practice often build upon or modify such systems. Furthermore, while our discussion will be grounded in language-model-

---

[2]I.e., they are not system-level interventions.

[3]See Section 7 for a comparison of agent infrastructure with adaptation interventions (Bernardi et al., 2024), which roughly include any intervention that is not a system-level intervention.

| Domain | System-level | Infrastructure |
|---|---|---|
| AI-agent safety | Fine-tuning, prompting, unlearning, filtering training data | Protocol to forget confidential information from an interaction, technologies to enable agent certification, distinct interaction interfaces for agents |
| Traffic safety | Driver training programs | Traffic lights, roundabouts, emergency lanes, camera-enforced speed limits |
| Health | Medication, exercise | Hospitals, emergency response systems, parks for exercise |

Table 2: We compare analogues to system-level interventions and agent infrastructure across different domains. These categories of interventions are often complementary.

based agents, the core ideas of agent infrastructure are largely architecture-agnostic and extend existing work across computational science, economics, and social science (Wooldridge, 2009; Perrier, 2025).

To further understand the distinctions between agent infrastructure and system-level interventions, consider traffic safety as an analogy. If we treat human drivers as analogous to AI agents, system-level interventions includes driver training programs. Infrastructure could include traffic lights, roundabouts, emergency lanes, and camera-enforced speed limits. We provide more comparisons in Table 2.

Just as the Internet relies on infrastructure like TCP (Eddy, 2022), HTTPS (Fielding et al., 2022), and BGP (Rekhter et al., 2006), we argue that agent infrastructure will likely be crucial for unlocking the benefits and managing the risks of agents. As an example of unlocking benefits, protocols that tie an agent's actions to a user could facilitate accountability and thereby reduce barriers to agent adoption. Analogously, the ability to perform secure financial transactions over HTTPS enables an e-commerce market in the trillions of dollars (Statista, 2024). As an example of managing risks, agent infrastructure can support system-level interventions. For instance, a certification system for agents could warn actors (e.g., other agents) not to interact with agents that lack certain safeguards, just as browsers flag non-HTTPS websites. In this way, agent infrastructure can use agents' interactions as a leverage point to improve safety: restricting an agent's interactions correspondingly restricts the agent's potential negative impacts.

This work identifies three functions that agent infrastructure could serve: 1) attributing actions, properties, and other information to specific agents or other actors; 2) shaping agents' interactions; and 3) detecting and remedying harmful actions from agents. We propose infrastructure that could help achieve each function, including analysis of use cases, adoption, limitations, and open questions. Our suggestions are primarily aimed at researchers and developers who may want to build agent infrastructure. These suggestions could also be useful to governments or funding bodies who may want to support its construction.

## 2 Agent Infrastructure

### 2.1 Categories of Agent Infrastructure

We focus on agents that interact with digital environments. Roughly speaking, we consider an agent to consist of the underlying machine-learning model and any primitives that provide basic functionality (e.g., memory, scaffolding). Language-model agents commonly act by issuing commands to tools, such as APIs that return the output of a separate software service (e.g., the weather).[4] Environments consist of such tools and any counterparties that interact with an agent (e.g., other agents, humans).

Some agent infrastructure applies to instances of agents. An **agent instance** is an instantiation of the underlying machine-learning model and any primitives for a particular user. Specifically, an instance would correspond to an interaction history, a set of available tools, memory, etc., much like process IDs in operating

---

[4]See Wooldridge (2009); Perrier & Lazar (2025) for further discussion of ontologies of agents.

systems correspond to running instances of programs.[5] Agent instances are a useful concept because different instances of the same underlying model and primitives can have different users or behave differently, and therefore potentially justify different treatment.

We identify three functions of agent infrastructure. **Attribution** is about attributing actions, properties, and other information to (instances of) agents, users, or other actors. **Interaction** is about shaping agents' interactions with the world. Finally, **response** is about detecting and remedying harmful actions from agents. To maintain a manageable scope, we do not discuss how infrastructure could help to manage more diffuse potential impacts, such as widespread unemployment. We summarize all the infrastructure we discuss in Table 1.

Infrastructure could be implemented both within a particular organization and between organizations or actors. Consider communication protocols as an example. In the first case, a business could use communication protocols to manage its agents internally, much as infrastructure like Slack and Google docs helps human employees work together. In the second case, a common communication protocol could help agents from different organizations communicate efficiently over the internet (Marro et al., 2024; Surapaneni et al., 2025). Adoption of agent infrastructure is likely more difficult in the second case, given that different actors could use mutually incompatible communication protocols (see Section 6.1). Our presentation of agent infrastructure is meant to cover both of these cases. For each piece of infrastructure that we present, we analyze how adoption could proceed at internet-wide scale.

## 2.2 Example

To illustrate how agent infrastructure could work, consider the task of running a company. In this hypothetical company, agents would perform or assist with the work that human employees would usually perform, such as operations, R&D, and legal tasks. Agent infrastructure could enable the following functions for this agent-run company.

- Human review: Human managers can make use of oversight layers (Section 4.2) to ensure that their agent employees are not malfunctioning.

- Coordination: A communication protocol (Section 4.3) and commitment devices (Section 4.4) can facilitate coordination between agents both within a company and from different companies. For example, agents could make commitments to carry out certain tasks, much as companies create contracts.

- Accountability: A lack of accountability mechanisms could discourage interaction with the company's agents. One way to address this issue is by ensuring that each agent has at least one human that can answer for the agent's actions (Section 3.1), and making this information available to those that interact with the company's agents or a third-party mediator.

- Undoing mistakes: Rollback mechanisms (Section 5.2) enable certain decisions, such as updates to compensation or internal data management processes, to be easily undone if an agent makes a mistake.[6]

## 2.3 Prioritization

We discuss some considerations for prioritization of certain types of agent infrastructure.

The effectiveness of certain infrastructure depends heavily on widespread adoption. For example, inter-agent communication protocols are only useful if both parties can use the same protocols. Agent IDs are also only useful if other parties can recognize them. On the other hand, oversight layers and rollbacks are immediately useful for addressing agent malfunction, even if few actors adopt them. For actors that are heavily resource-constrained, such considerations could suggest focusing on infrastructure that does not require widespread adoption.

---

[5]An instance is an abstraction and may not correspond to a hardware separation. For example, inference calls for different instances could be batched together on the same piece of hardware.

[6]The conditions under which a decision should be undone is a separate question. Rollback mechanisms merely provide flexibility.

Some infrastructure is only useful once agents obtain certain capabilities. For example, commitment devices are most useful when agents are capable of negotiating and carrying out complex agreements. Infrastructure for capabilities that are further off could be delayed.

Different types of infrastructure are useful for different threat models. For example, inter-agent communication protocols and commitment devices are most important for avoiding cooperation problems between agents (Hammond et al., 2025). Oversight layers and rollbacks are most useful for cases of misuse or misalignment. Other infrastructure, such as identity binding and agent IDs, can both facilitate cooperation and help to address harms.

We take no explicit stance on prioritization and leave further analysis to future work.

## 3 Attribution

Attribution infrastructure attributes actions, properties, and other information to (instances of) agents, users, or other actors. **Identity binding** links an agent or its actions to an existing legal identity. **Certification** would provide assurance about behaviour and properties of an agent instance. An **agent ID** would be a container of information about an agent instance, such as associated certifications. In Figure 2, we depict how these pieces of infrastructure fit together.

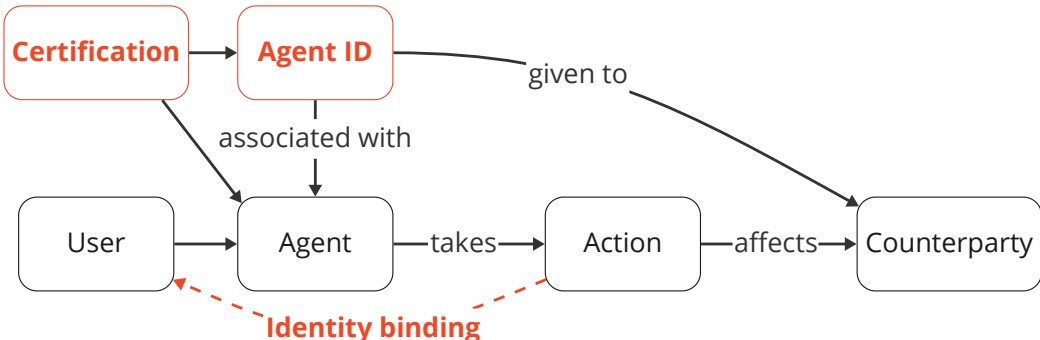

Figure 2: **Identity binding** links an agent or its actions to an existing legal identity, **certification** would provide assurance about behaviour and properties of an agent instance, and an **agent IDs** would be a container of information about an agent instance. We illustrate **identity binding** for a user, but an agent could also be bound to other actors, such as those involved in overseeing it

### 3.1 Identity Binding

**Description**: AI agents are not legal entities[7]. The ability (but not necessarily the requirement) to associate an agent's actions with a legal entity, such as a human or a corporation, could help existing institutions accommodate agents. For example, an agent's VOIP calls could be tagged with the identity of the person controlling the agent, so as to enhance accountability for agent-enabled scams (Fang et al., 2024b). Analogously, cars are associated to drivers through license plates and registries. Such binding could be helpful even when the relevant legal entity is not fully responsible for all of the agents' actions. For example, in addition to the employee using an agent, managers reviewing the agent's actions and the corporation hiring the employee could potentially bear some responsibility. With this motivation, we define **identity binding** to be a process that consists of (1) authentication of an identity and (2) linking that identity to an (instance of an) agent or its actions. Crucially, linking an identity to an agent does not mean making the identity publicly visible. For example, identity information could be accessible only to authorized parties, such as if needed for legal action, or only reveal limited information, such as the fact that a real person is responsible for

---

[7]Some argue that sufficiently capable future agents should have legal rights (Salib & Goldstein, 2024).

the agent (Adler et al., 2024). Further below, we discuss how existing infrastructure could help accomplish (1) and (2), along with modifications that may be needed.

**Potential functions**:

- Accountability: Malicious uses of agents could be harder to address if all agents acted anonymously. For example, anonymity enables Sybil attacks, wherein someone creates multiple fake identities to undermine a network (e.g., an online voting system).

- Trust: Counterparties could be more willing to engage in productive interactions with agents that are bound to legal entities because it could be easier to obtain recourse in case of harm. Analogously, identity verification of drivers reassures users that would otherwise be dissuaded from ridesharing by personal safety considerations.

- Application of laws: Identity binding could allow the broader application of existing laws to interactions between agents. For example, at least as a starting point, identity binding could provide evidence that an agent did indeed enter into a contract on behalf of a particular principal (Benthall & Shekman, 2023; Aguirre et al., 2020).

**Existing alternatives**: Existing technologies can provide some functionality for authentication and identity linking, but have limitations. User authentication is already required for many digital services, such as banking (through know-your-customer regulations) or services requiring an account (through logging in with OAuth). The actions of agents interacting with such services would by default be attributable to the user. However, some services only provide attribution to a pseudonymous internet user, rather than a specific legal entity. In addition, there is currently no comparable authentication system for agent-agent interactions or actors other than users. The former may be important for agents that are significant economic actors, whereas the latter could include actors that assume responsibility for some of the agent's actions (e.g., an insurance company) or other overseers of the agent.

Identity information could be linked to an agent in a variety of ways. Drawing from work on content provenance (e.g., (C2PA, 2023)), identity information can be included as metadata or as a watermark. However, metadata can be stripped from content and watermarks can potentially be removed (Zhang et al., 2023). Furthermore, privacy considerations could weigh against revealing identity information in this way. If so, a potential alternative could be to establish a trusted intermediary that would hold this identity information (Kling et al., 1999). Counterparties could request that this intermediary reveal the information or use it appropriately in the event it is needed. As an analogy, ISPs can share information about their users in response to a criminal investigation. It remains to be seen how such a function could be provided for agents.

**Adoption**: Digital platforms are likely to be crucial for the widespread adoption of identity binding. First, users might be more likely to engage with platforms that include or require identity binding. Some evidence suggests that contractors use AI systems on platforms like Upwork (Veselovsky et al., 2023). Dharmesh (2024) is also creating a service where AI agents play the role of crowdworkers and are matched to tasks. In both of these cases, customers are likely to engage more fully if they know there is a person ultimately responsible for the service, similar to identity verification for gig economy services (Uber, 2025; DoorDash, 2025). Second, platforms might require identity binding to avoid Sybil-like attacks. For example, an individual recently defrauded Spotify of millions of dollars by synthetically generating music and creating multiple fake accounts to stream it (U.S. Attorney's Office, 2024). Third, identity binding could be a natural extension existing single sign-on practices. For example, a Google account can already be used to log on to many non-Google-owned different websites and services.

**Limitations**: Identity binding entails at least some collection and use of user information, even if that information is only shared with authorized actors when needed. Security vulnerabilities analogous to those in existing authentication systems could compromise user information (Batra et al., 2024; Microsoft Threat Intelligence, 2023). More broadly, reduced anonymity (or the threat of reduced anonymity) could threaten personal safety (e.g., doxxing), hinder whistleblowing, and reduce expression of novel ideas (Kling et al., 1999; Zittrain, 2006; Choi, 2012; Kang et al., 2013). Such concerns imply that identity binding may not be appropriate in every application of agents, and proportionate confidentiality provisions will be necessary when

it is used. Analogously, although it can be abused, legislatures and courts have recognized the importance of identification of internet users under certain conditions (Ekstrand, 2003; Choi, 2012; noa, 2018).

**Research questions**:

- What kinds of actions can and should be linked to identities? For example, it is likely infeasible to link mouse clicks to agents. Would it be preferable to link groups of actions instead?

- What key applications of agents could identity binding enable?

- How can identity binding build upon existing digital identity solutions, such as (national) digital wallets or single sign-on?

- If intermediaries hold identity information, who should be able to request access to or use of the information, such as for obtaining recourse? Under what conditions should such access or use be granted?

### 3.2 Certification

**Description**: Before interacting with an agent (instance), counterparties might want assurances about its operation, behaviour, or properties. For example, agents whose actions are regularly reviewed could be more trustworthy than agents who do not undergo such review. Claims about operation, behavior, or properties could come from a user, an agent deployer, or other actors (e.g., third-party auditors). Certificates including such claims could be associated with the relevant agent and made visible to counterparties interacting with it. Analagously, SSL certificates make claims to website users about domain ownership. **Certification infrastructure** consists of tools for making, verifying, and revoking certificates for agents. Verification includes how those viewing a certificate can verify both its claims and that it corresponds to the agent with which they are interacting.

**Example claims**: We provide potential example claims of an agent's operation, behavior, or properties that could be included in a certificate.

- The tools the agent can access: Certain tools, such as commitment devices (Sun et al., 2023), could make agents more reliable or useful.

- Capabilities, accounting for post-training modifications: Weaker agents could be more susceptible to prompt injections from stronger agents (e.g., stronger agents could be more capable at detecting such attempts). If so, a user might want their agent only to interact with agents that are at most as strong as their agent. Accounting for post-training modifications would likely involve analysis of an agent's behaviour log. Doing so in a privacy-preserving way is an open problem (Trask et al., 2023; 2024).

- The level of autonomy with which the agent is authorized to act: Agents that act with more regular human oversight could be, but are not necessarily (Parasuraman & Riley, 1997; Skitka et al., 1999; Green, 2022), more trustworthy.

- How an agent handles sensitive information: For example, an agent could follow a protocol to delete sensitive information from context and memory after use (Rahaman et al., 2024).

**Potential functions**:

- Trust: Certificates for safety-relevant properties can build trust in agent interactions. For such trust not to be misplaced, a certificate should at minimum only contain claims about what can feasibly be known about the agent, given the state of the science. For example, it may be difficult to provide useful bounds on whether an agent will ever violate a safety constraint.

- Races to the top: If counterparties can selectively interact with agents that have certificates for pro-social properties (e.g., that use cooperation mechanisms (Sun et al., 2023)), developers and users could be incentivized to respectively develop and use agents with those properties.

- Supporting recourse: Certain forms of recourse (e.g., undoing an action; see Section 5.2) could rely on an auditor to certify that, for example, an agent was prompt injected.

**Existing alternatives**: Auditing of companies involved in the development or use of agents, such as developers or cloud compute providers, can be a component of certification of agents. For example, an auditor could verify that a cloud compute provider has policies for only running agents that have passed certain safety standards. Yet, certifying such companies could be ineffective if large numbers of users modify and deploy their own agents.

In the absence of any additional certification infrastructure, developers, deployers, and users could make their own claims about their agents. However, counterparties may be unable to assess the veracity of such claims. Evaluating AI systems is not straightforward and empirical experiments can be subtly misleading (Leech et al., 2024). Even so, entities may still be able to assess the quality of an agent's actions and reject them if necessary. For example, a counterparty may not know in general how reliable a software engineering agent is, but could run software to verify the correctness of the agent's code.

**Adoption**: Demand for certification is likely to vary across domains and contexts. For instance, protocols for forgetting sensitive information can enable agreements that would otherwise not be possible (Rahaman et al., 2024). Certifying that an agent can follow such protocol seems more important for high-stakes negotiations. Feasibility and economic cost could also affect demand. For example, properties that require access to interaction logs (e.g., whether a prompt injection is present) could raise privacy concerns, in the absence of privacy-preserving certification methods (Trask et al., 2023; 2024). That said, demand for certification could be high for agent use within government and regulated areas (e.g., health). Analogously, software products in such domains often have to meet quality standards (e.g., Health and Human Services Department & Food and Drug Administration (1996)).

Finally, as discussed above, developers and deployers could support certification as a way to credibly signal the trustworthiness of their agents.

**Limitations**: It could be impossible or extremely difficult to verify certain properties. Providing meaningful guarantees about the behavior of a system in general is an active area of research (Dalrymple et al., 2024). If meaningful guarantees are not technically possible, certifications could provide a false sense of security. Other properties could be feasible, but potentially impractical or invasive, to certify. A potential example could be whether an agent has interacted with a particular, other agent.

Furthermore, actors creating certificates for agents could be incompetent, captured, or fail to perform any useful certification. A notable case is TrustARC (formerly TRUSTe), who provided privacy certification for websites. Although TrustARC claimed to verify website privacy policies, it often did not conduct any such checks whatsoever (Connolly et al., 2014). Surprisingly, as an example of adverse selection, Edelman (2009) provides evidence that websites that obtained TrustARC certification were significantly less trustworthy than websites without it. potential Ways to address this problem include oversight over actors performing certification of agents and transparency requirements about their processes.

**Research questions**:

- For the example claims we list above, who should be making the claims? How could the claims be verified?

- What other claims about an agent would be useful to make?

- How can actors verify that a certificate corresponds to the agent instance with which they are interacting?

- Under what conditions should a certificate be revoked? For example, claims about an agent's future behavior could become outdated the longer the agent operates.

- How would certification of agents interact with existing auditing practices and regimes (Raji et al., 2020; 2022)?

- How could existing infrastructure (e.g., verifiable credentials (Sporny et al., 2024)) support certification of agents?

- What existing certification methods in information systems and cybersecurity could provide useful inspiration (Oberkampf & Roy, 2010; Perrier, 2025; Perrier & Lazar, 2025)?

- What oversight or transparency requirements should exist for actors providing certificates for agents?

- What recourse should be available if certificates contain mistakes or are otherwise compromised in some way?

### 3.3 Agent IDs

**Description**: An **agent ID** would at minimum consist of a unique identifier[8] for an agent instance, and potentially include other information relevant to the instance, such as a system card or certifications (Chan et al., 2024b). ID-like systems exist across many domains to facilitate identification, such as serial numbers on consumer products, tail numbers on aircraft, registration numbers for businesses, and URLs/URIs to locate resources on the internet. Although any AI system could have an ID, IDs could be significantly more useful for agents because they take actions in the world (see below).

**Potential functions**: Many of these uses draw from Chan et al. (2024b).

- Supporting certification: An ID could present an agent's certifications (Section 3.2) and bound identities (Section 3.1) to entities that seek to engage with it.

- Incident response: IDs could provide information that is useful for incident response. For example, an agent could create sub-agents to split up a harmful task into multiple sub-tasks (Jones et al., 2024). An ID could link the activities of these sub-agents to the original agent. An ID could also be useful for detecting cross-platform incidents, where an agent harms multiple counterparties.

- Enabling targeted action against particular agents: Much like how process IDs in operating systems enable monitoring and resource restriction, an ID could enable interventions on the agents that are causing harm.

**Existing alternatives**: For agents that interact with digital services, OAuth tokens or (read-only) API keys could aid incident attribution. However, such tools do not apply to interactions between agents. These tools are also service-specific. For example, an OAuth token cannot link the activity of an agent across different counterparties, nor can it link the activities of sub-agents to the agent that created them.

IP addresses could also help incident attribution, but may not reliably identify an agent instance. Indeed, they may not be static and can also be shared amongst different internet users.

**Adoption**: Identifiers could be a part of existing obligations to inform humans when they are interacting with AI systems (noa, 2024, Art. 50(1)). As agents are assigned increasingly complex roles, counterparties could demand more information from IDs so as to ensure reliable interactions (e.g., through certification) and facilitate incident attribution.

Counterparties that expect only to interact with agents (or which only interact with agents through specific channels [Section 4.1]) could require IDs before an interaction. Other counterparties which also interact with systems other than agents may have more difficulty doing so. For example, suppose a counterparty engages with both agents and traditional software programs. Extending an ID system from agents to all software programs could be infeasible, since traditional software programs do not have IDs. A potential stopgap measure could be CAPTCHA-like methods (Tatoris et al., 2025). The effectiveness of this measure depends upon whether agent behaviour is distinguishable from human and other bot behavior, such as through adversarial vulnerabilities.

**Limitations**: IDs could allow attackers to target particular agents and their users. If activity associated with an ID is logged, leaks of such information could allow attackers to identify high-value targets. Knowing an agent's frequently used services could also facilitate exploits like prompt injections (e.g., attackers could infect particular websites). When interacting with a particular agent, attackers could exfiltrate information or jailbreak the agent to act on the attacker's behalf, such as by performing financial transfers.

IDs could also be stolen or otherwise compromised. For example, IDs that depend on certificate authorities are vulnerable to existing security vulnerabilities (Berkowsky & Hayajneh, 2017).

---

[8]A cryptographic scheme is necessary to prevent spoofing of the identifier. See Chan et al. (2024b) for more details.

**Research questions**:

- How could IDs complement existing tools for incident investigation?

- Where should IDs be required, and what information should an ID include?

- How can counterparties incentivize or enforce ID usage (e.g., by making sure that agents use unique channels (Section 4.1))?

- When should IDs link to information identifying the user?

- What should entities be allowed to do with an identifier? For example, should an entity with an identifier be allowed to contact the associated agent, or otherwise interact with it?

- What other systems need to be in place to make IDs more useful? E.g., should there be a registry of certain agent IDs or a reviewing system?

- How can the design of IDs avoid known problems with existing Internet infrastructure (e.g., certificate authorities) (Berkowsky & Hayajneh, 2017; Dumitrescu & Pouwelse, 2024)?

- How can the design of IDs limit abuse (e.g., attackers use IDs to target particular agents and their users)?

## 4 Interaction

Interaction infrastructure consists of protocols and affordances to shape how agents interact with their environments. **Agent channels** would isolate agent traffic from other traffic. **Oversight layers** would enable actors to intervene on an agent's behaviour. **Inter-agent communication** protocols would help to facilitate joint activities amongst groups of agents. **Commitment devices** are mechanisms that enforce commitments between agents. We depict these pieces of infrastructure in Figure 3.

### 4.1 Agent Channels

**Description**: For interactions with a digital service (e.g., payment services), an **agent channel** would separate agent traffic (or AI traffic in general) from other digital traffic. An agent channel could, for example, facilitate incident management by temporarily suspending access if a language-model worm spreads (Cohen et al., 2024; Gu et al., 2024). Traffic separation is common across many applications. For example, digital payments companies isolate systems that handle cardholder data (Council, 2016). Some hospitals separate networks for storing and using different kinds of medical data, so as to safeguard privacy in case one network is compromised (Digital, 2023). Wi-Fi routers provide guest networks that restrict the activity of unfamiliar devices.

There are two axes of design considerations for agent channels. First, agent channels could be different software interfaces or protocols for agents, or could take the form of more dedicated internet infrastructure. As examples of the former, there are efforts to build agent-specific web browsers (Turow, 2024) and interaction protocols (Kaliski, 2024; Anthropic, 2024b). As examples of the latter, agents could have a dedicated block of IP addresses (which could perhaps function as IDs Section 3.3), or separate routing systems could handle agent traffic.

The second axis is how to operationalize traffic separation. Rate limits on human channels and the relative absence or weakening of such limits on agent channels could incentivize the use of the latter. However, it may in practice be difficult to separate agent or AI traffic from software traffic in general. For example, agents could write software that interacts with counterparties.[9]

**Potential functions**:

- Monitoring: Separate transaction data for agents, for example, can help to understand the potential macroeconomic effects of high volumes of agent activity (Van Loo, 2019).

---

[9]If agents come to mediate large fractions of digital interactions, there may be very little software that is not related to an agent in some way.

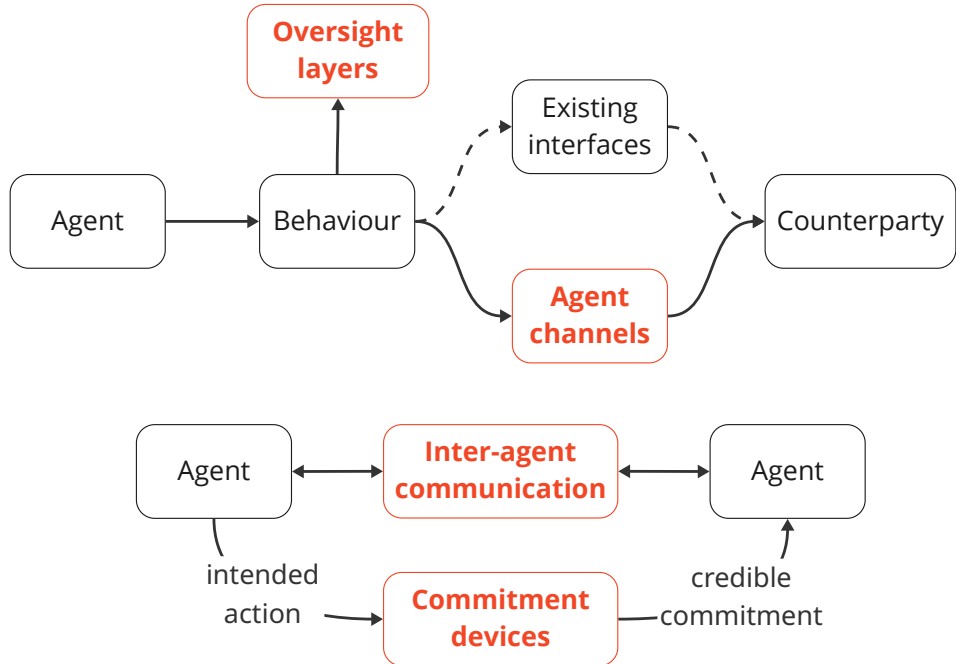

Figure 3: **Agent channels** separate agent traffic from other digital traffic. **Oversight layers** enable actors to intervene on an agent's behaviour. **Inter-agent communication** help to facilitate joint activities amongst groups of agents. **Commitment devices** enforce commitments between agents. See Section 4 for discussion.

- Managing incidents: Isolation of traffic could help to manage incidents. For example, providers of services could shut down agent channels, but not human channels, to restrict spread of a text or image worm (Cohen et al., 2024; Gu et al., 2024).

- Enforcing agent-specific rules: Agent channels could implement rules to support reliable agent use, such as around rolling back unintended actions (Section 5.2) or how to handle sensitive data.

- Reduced attack surface: Human graphical interfaces have a large attack surface. For example, imperceptible changes to a website page could hide a prompt injection. APIs designed for agents could reduce this attack surface, although text-based prompt injections would still be possible.

- Supporting IDs: As discussed in Section 3.3, agent channels could make it more feasible to require IDs from agents.

**Existing alternatives**: Agents currently use existing software interfaces when interacting with web services (Significant-Gravitas, 2024). In the absence of dedicated channels, behavioural tests could try to identify and isolate agent behaviour. CAPTCHAs tend to be unreliable (Hossen et al., 2020; Hossen & Hei, 2021; Searles et al., 2023). Another potential test is whether an entity interacts with an interface at faster-than-average human speeds. If so, the entity could undergo more intensive monitoring or have more limited access to functions, under the suspicion that it might not be human. This differential treatment would effectively constitute an agent channel, without building specific interfaces or other dedicated internet infrastructure.

**Adoption**: Agents currently have difficulty interacting with human interfaces (Furuta et al., 2024; Tao et al., 2023; Gur et al., 2024; Xie et al., 2024). If agent channels are optimized to help agents function more efficiently and reliably (Song et al., 2024), such as by simplifying interfaces (e.g., website navigation instructions) or imposing fewer limitations (e.g., higher rate limits, reduced latency) compared to human interfaces, there could be much demand for them. Yet, agents could soon become capable of interacting with with human interfaces. Encouraging the use of agent channels could therefore be time-sensitive.

**Limitations**: Agent channels could fail to cover enough agents or could cover entities besides agents. On the first point, agent channels would likely be less effective if only a small number of agents use them. Monitoring of agent channels may miss incidents, and interventions on agent channels may not catch all agent activity. On the second point, although it seems difficult to prevent other entities from using agent channels, this limitation may not be significant. For example, any rules applied to agent channels (e.g., slowing the spread of a language-model worm by restricting activities) would simply also apply to those entities.

**Research questions**:

- How should agent channels be designed and implemented? Can their creation be automated (e.g., automatic API creation)?

- What could encourage the adoption of agent channels?

- Is it possible (or desirable) to isolate agent traffic for functionalities already provided by traditional software interfaces (e.g., email (Significant-Gravitas, 2024), weather retrieval (OpenAI, 2024))?

- What policies should govern the use of agent channels to influence or constrain agents?

## 4.2 Oversight Layers

**Description**: The potential for unreliable agent behaviour motivates review of the agent's actions. An **oversight layer** would consist of (1) a monitoring system to detect when external intervention is required during an agent's operation and (2) an interface for an actor to intervene. Examples of external intervention include the provision of additional information (e.g., about the task or what the agent is not authorized to do) or rejection of the agent's action. The actor in (2) could be a human user, but not necessarily. For example, a trusted, automated system could take the place of a user (Greenblatt et al., 2024; Dalrymple et al., 2024), especially if the oversight layer flags many actions. An oversight layer could also be useful for *any* actor that engages with an agent, not just a user. For example, a manager in a company could use an oversight layer to review actions from agents used by individual employees.

**Potential functions**:

- Rejecting unsafe actions: Agents could take consequential actions that are unbeknownst to, or unintended by, their users. For example, a malicious actor could prompt-inject an agent to drain a user's bank account. Similar to credit card fraud detection, methods for detecting when actions may be unintended could protect users.

- Improving functionality: As Hewett (2024) discusses, an oversight layer could ask a human for help whenever an agent is incapable of completing a particular action (e.g., misinterprets the task or is stuck in a loop), without the human having to restart the agent from scratch.

- Accountability: An oversight layer could make it easier to assess responsibility for an agent's actions, such as by recording the explicit approval of a user for an action that harmed another party.

- Generating useful information: Oversight layers could generate useful information about which agents tend to take actions that overseers reject more often. This information could enable, for example, insurance to be designed around the expected reliability of an agent.

**Existing alternatives**: Some technical frameworks for agents already include some kind of oversight layer (Chase, 2022; Wu et al., 2023). See below for additional research questions to extend upon this work.

Analogues to oversight layers are common across many settings. In banking, certain activity can trigger account freezes or requests for further review. For many digital services, logging in from an unfamiliar device triggers a request for user approval. In cyber security, endpoint detection and response tools monitor for and flag suspicious activity. Oversight layers for agents could build upon or integrate with such existing technologies.

**Adoption**: There could be strong incentives to develop and implement oversight layers because they make agents more usable. However, it is unclear if the market will provide them to a socially optimal degree (or what a socially optimal degree would be). Analogously, although more granular social feed controls (e.g., reducing the degree of sensationalist content) would make social media more usable, major social media companies have not provided such options. Indeed, some controls may be against the financial interests of such companies. For example, removing sensationalist content could reduce engagement and ad revenue. Analogously, AI companies could hesitate about providing users with the ability to limit how agents engage with activities that could be profitable for the company, such as making financial transactions or viewing advertisements.

**Limitations**: Active attention is required if oversight layers flag potentially problematic behaviour. Humans could ignore such flags if they are too frequent or present information in a complex manner. Analogously, internet users rarely read terms of service. Humans could also have predispositions to trust automated systems without sufficient verification of their behaviour (Parasuraman & Riley, 1997; Skitka et al., 1999; Goddard et al., 2012). At the same time, AI systems could eventually be better than humans at catching problematic behaviour. For instance, language models can already help catch vulnerabilities in real-world code (Big Sleep Team, 2024). However, using more capable AI systems for monitoring incurs inference costs. Using less capable systems, or simply decreasing inference budgets, could reduce the effectiveness of monitoring. Further study is needed to understand whether the default equilibrium will be desirable.

**Research questions**:

- Who should be able to access information generated by oversight layers?

- How can an oversight layer gracefully interrupt an agent when it requires external intervention and integrate additional information or actions from another actor?

- How can we design interfaces that allow users (or other actors) to set the degree of oversight (e.g., how often the user is notified of a flagged action)?

- How can we build trusted, automated systems (Greenblatt et al., 2024; Dalrymple et al., 2024) that can intervene so as to minimize the degree of human intervention required?

- Given trade-offs between effectiveness and economic cost, will the market sufficiently provide for oversight layers? If not, what government action can best incentivize their development and implementation?

### 4.3 Inter-Agent Communication

**Description**: Agents will likely interact not only with existing digital services, but also with other agents. For example, agents from different users could collaborate on shared projects. **Inter-agent communication infrastructure** includes (1) rules for how agents communicate and (2) the underlying technical systems to enable such communication. On (1), in addition to point-to-point communications, infrastructure may also need to facilitate broadcast-like communications (e.g., for important announcements). On (2), technical systems include an addressing system and the underlying messaging protocol. Agent channels (Section 4.1) are similar to inter-agent communication infrastructure, but focuses on interaction with existing digital services rather than interactions between agents.

**Potential functions**:

- Notification: Communications can convey important information. For example, an agent could warn other agents about the discovery of a security vulnerability.

- Cooperation: Communications could be useful for coordinating the activities of different agents, whether for completing shared projects or helping to resolve collective action problems.

- Negotiation: Agents could use communications to negotiate on rules for an interaction (Marro et al., 2024), such as how to handle confidential information (Rahaman et al., 2024).

**Existing alternatives**: Some existing communication infrastructure may be difficult for agents to use because they require human involvement. For example, signing up for a Gmail account requires access to a phone number. It may also be difficult to add agent-specific functionality onto existing communication infrastructure. For instance, Facebook Messenger does not currently support broadcasting functions.

On the other hand, existing open protocols could be the foundation of more specific agent protocols. For example, Marro et al. (2024) describe a meta protocol wherein LLM agents can negotiate, in natural language, the subsequent use of more efficient, structured data formats (e.g. JSON). As another example, Google's A2A protocol uses HTTP for transport between client and server agents (Surapaneni et al., 2025).

**Adoption**: At a high level, deficiencies in existing communication protocols could spur the adoption of more specialized alternatives for agents. As discussed above, required human intervention for existing communication infrastructure could slow or inconvenience interactions. On the other hand, since many agents are able to communicate in natural language, newer protocols could take advantage of this flexibility (Marro et al., 2024).

As we discuss further in Section 6.1, network effects are likely to be important for the adoption of specific communication protocols as common standards. Buy-in from agent developers and deployers, digital platforms, and large organizations that use agents is likely to be crucial. For example, Google is collaborating with the number of large enterprises for its A2A protocol (Surapaneni et al., 2025).

**Limitations**: As with any communication infrastructure, inter-agent communication infrastructure could be abused. For example, broadcast functionalities could be sources of spam. Scammers could use an addressing system (e.g., a "phone number" or "URL" for each agent) to target particular agents. Worms that target the adversarial vulnerabilities of AI agents (Cohen et al., 2024) could spread along communication channels. The severity of such problems will likely depend the development of countermeasures, such as whether particular agents can be blocked. Analogously, email scanners can help block malicious payloads and social engineering attempts.

More broadly, organizations could build communication protocols (and infrastructure in general) in ways that are beneficial to their financial interests but are detrimental to other parties. For example, ad-supported businesses could prefer communication protocols that make it difficult for agents to block advertising from other agents. While advertising may not be undesirable in and of itself, the dominance of ad-supported businesses could make it difficult for alternative business models to arise (Lazar et al., 2024).

**Research questions**:

- Beyond encryption, how could inter-agent communication protocols be secured? For example, is there a way of making sure that messages do not contain jailbreaks or steganographic communications?

- What are use cases for broadcasting, and how should it be designed?

- What other functionalities should inter-agent communication infrastructure include? For example, a communication protocol could include a way to signal or carry out commitments (Section 4.4).

- How can inter-agent communication infrastructure integrate with other types of agent infrastructure, such as oversight layers (Section 4.2), agent IDs (Section 3.3), and identity binding (Section 3.1)?

## 4.4 Commitment Devices

**Description**: An inability to credibly commit to particular actions is a key reason why rational actors sometimes do not obtain cooperative outcomes (Fearon, 1995; Powell, 2006; Clifton et al., 2022). Commitment devices—mechanisms that can enforce commitments—aim to address this issue. Examples of commitment devices include escrow payments, assurance contracts, and legal contracts. As agents become increasingly important economic and social actors, commitment devices designed for agents can help to achieve positive societal outcomes. Such commitment devices could simply be interfaces connecting agents to existing human commitment devices, or could be specially designed for agents.

**Potential functions**:

- Funding productive activities: The funding of productive activities can suffer from collective action problems (e.g., underprovisioning of public goods). Commitment devices could help agents with economic resources, whether used independently or on behalf of a user, cooperate to fund such activities. For example, future agents could initiate a vast array of projects or companies. Analogues to Kickstarter (i.e., assurance contracts (Bagnoli & Lipman, 1989; Tabarrok, 1998)) could facilitate funding of such projects or companies.

- Helping to avoid the tragedy of the commons: Commitment device could allow agents to agree not to carry out risky activities as long as others do the same, even if such activities advance the interests of the individual agent or user. For example, competitive pressures in AI development likely lead to underinvestment in safety (Askell et al., 2019). Future agents that are able to perform such development (Wijk et al., 2024) or run AI companies could commit, for example, not to build systems that surpass certain capability thresholds without sufficient safety guardrails, so long as others do the same.

**Existing alternatives**: Many existing commitment devices depend upon (threats of) action in the physical world or human social norms, such as legal contracts or making public announcements. These threats or norms can be communicated to language-model-based agents. However, it is not clear how reliably such agents will respond. As one piece of evidence, Li et al. (2023) find that emotional cues can affect language model performance. On the other hand, language models can still fail to reliably follow instructions (Baker et al., 2025).

Software-based commitment devices, such as smart contracts, are likely to be more fruitful in the short term. For example, agent $A$ could commit to transfer an amount under certain conditions to agent $B$ simply by writing a smart contract and deploying it onto the blockchain, which lets agent $B$ verify the contract. This use case is no different from if two humans want to commit to the same agreement. In this sense, agent-specific modifications to the core technology of smart contracts are not necessarily needed. Rather, the main limitation is the types of transactions that can be carried out on a blockchain. Many agreements involve an information source in the real world. For example, an agreement about a building might involve terms related to its appearance or other physical conditions. Without a reliable digital connection to such sources, smart contracts are inapplicable.

**Adoption**: Since commitment devices facilitate agreements, there is a market incentive to build and adopt commitment devices. Analogously, many private companies provide escrow services. Kickstarter is an example of an assurance contract.

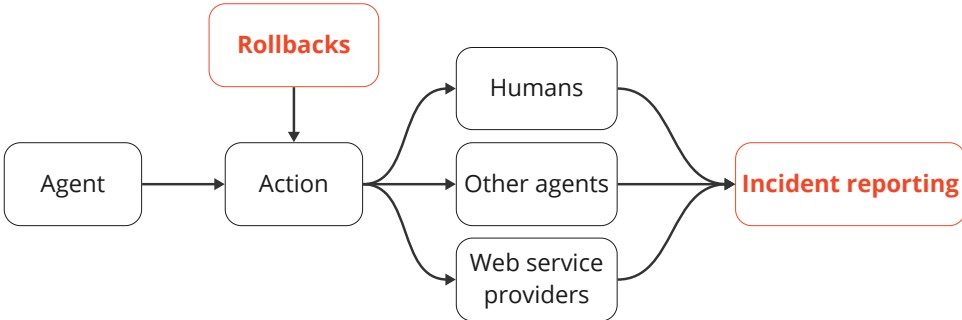

Figure 4: **Incident reporting** systems would collect information about and respond to events that could result in harm, while **Rollbacks** are tools that would void or undo an agent's actions. See Section 5 for further discussion.

Adoption of commitment devices that depend upon a blockchain could face barriers. One potential barrier is simply not many types of transactions cannot yet be carried out on a blockchain, as discussed above. Another barrier is potential lack of consumer trust in blockchain technologies, given high-profile scandals (Sherman & Hoskins, 2023).

**Limitations**: A commitment device is only useful in so far as it is employed. Depending on how a commitment device is implemented, users could be able to circumvent them. For example, if a commitment device depends upon modifying an agent's weights so that it takes an action, the user could simply shut the agent down.

Commitments could also lead to negative outcomes. For example, an agent that misunderstands a user's preferences could mistakenly commit funds to a project that the user does not support. More broadly, improved cooperation between agents could come at the expense of actors not included within the cooperative coalition (Dafoe et al., 2021; Fish et al., 2024), such as humans who are less capable of using cooperative technologies.

**Research questions**:

- How can adoption of software-based commitment devices, such as smart contracts, be improved?

- How could commitment devices integrate with inter-agent communication channels (Section 4.3)?

- When should commitments between agents be legally enforceable?

- Will private actors sufficiently provide for commitment devices?

- If future agents are responsive to legal incentives,[10] what other commitment devices would still be necessary?

## 5 Response

Response infrastructure consists of tools for detecting and remediating harmful actions from agents. **Incident reporting** systems would collect information about and respond to events that could result in harm. **Rollbacks** are tools that would void or undo an agent's actions.

### 5.1 Incident Reporting

**Description**: As agents become more involved in economic and social activities, they could both cause or witness events that could result in harm. Responding to and learning from such events, which we call

---

[10]One possibility is to directly train agents that follow the law (O'Keefe et al., 2025)

**incidents** (Wei & Heim, 2024), will require information gathering. Many compute providers and AI deployers already monitor their AI systems (Cloud, 2023; ChrisHMSFT et al., 2023; OpenAI, 2023b; AWS, 2023; OpenAI, 2023c; DeepMind, 2024; Anthropic, 2024c), but the downstream impacts of agents could be invisible to such actors. For example, a deployer of agent $A$ would not observe the impact of any infectious code or prompts given to agent $B$. On the other hand, if agents become involved in many interactions, they could become a valuable source of information about incidents involving other agents. **Incident reporting** infrastructure consists of (1) tools and processes for agents and other actors to collect information about incidents and file incident reports, (2) ways to filter for informationally valuable reports and prevent inundations of spurious reports (Schneier & Sanders, 2024), and (3) scalable methods for analyzing and responding to reports. Such infrastructure could help prevent future incidents by informing improvements to agent systems and safeguards.

Others have also proposed incident reporting systems, but without the focus on agents as a source of information (Costanza-Chock et al., 2022; Wei & Heim, 2024). More broadly, analogous systems include financial whistleblower programs for financial institutions (Securities and Exchange Commission, 2024), public reporting systems for vehicle offenses (Government of the United Kingdom, 2024), and cybersecurity incident (Government of the United Kingdom, 2022) or vulnerabilities (MITRE Corporation, 2025) reporting. Another related idea is using weaker AI systems to supervise stronger AI systems (Greenblatt et al., 2024).

**Potential functions**:

- Improving safety practices: Incident reports could surface novel harms or impacts that upstream actors, such as regulators or developers, had not considered. These actors can incorporate such insights into safety practices, such as by improving their monitoring systems or development processes.

- Monitoring locally run agents: No intermediaries can monitor the activities of agents that are locally run. Incident reports from counterparties that interact with these agents could help to fill this gap.

**Existing alternatives**: Existing incident reporting systems for AI are limited. Some AI companies run bug bounty programs (Anthropic, 2024a; OpenAI, 2023a). OpenAI allows users to flag whether a GPT is illegal or fails to function (OpenAI, 2024). Although some civil society organizations run reporting systems that accept broader types of incidents (AI Risk and Vulnerability Alliance, 2024; AIAAIC, 2024; Responsible AI Collaborative, 2024), no mechanisms yet commit government or industry to respond to those incidents. Furthermore, existing programs are designed to collect information only from humans, not AI agents.

A potential challenge for existing incident report systems in other countries is scaling up resources to detect and block spurious reports, as agents become more capable of crafting compelling reports (Schneier & Sanders, 2024). Identity binding (Section 3.1) could potentially alleviate the risk of spurious reports.

**Adoption**: Incident reports could help companies improve product functionality. Yet, incident reports could also result in reputational damage, such as being associated with building unsafe AI agents. Confidential reporting systems, such as in aviation, could reduce this hesitation so as to improve safety learning across an industry. On the other hand, ensuring accountability for incidents will likely require some government intervention.

**Limitations**: For interactions with anonymous agents, it is unclear if a counterparty would have information that could facilitate further investigation. To that extent, counterparties could prefer interacting with agents that have an ID (Section 3.3) or for which identity binding (Section 3.1) is possible. Attackers could also abuse incident reporting systems, for example to distract regulators from real incidents (Schneier & Sanders, 2024).

**Research questions**:

- If an agent causes an incident, what information could feasibly be included in an incident report? For example, an agent ID (Section 3.3) could be included if it is available.

- How can existing, domain-specific incident reporting systems make filing reports more accessible to agents? How can operators of incident reporting systems manage influxes of spurious reports? For example, could incident reporting systems require personhood credentials (Adler et al., 2024)?

- Which organizations should respond to incidents involving agents run on local compute?

- How can agents help with managing and responding to incident reports?

## 5.2 Rollbacks

**Description**: Voiding or undoing an agent's actions could be useful in certain circumstances. For example, if agents often malfunction and make incorrect financial transactions, reversal of such actions could reduce harms to users. **Rollback infrastructure** would enable voiding or undoing an agent's actions. It includes both mechanisms to implement such a voiding or undoing action, as well as interfaces for actors to perform or request it. As an example of the former, Patil et al. (2024) build a LM runtime that allows reversal to a previously saved state.

**Potential functions**:

- Undoing unintended actions: Agents that malfunction or are hijacked (Greshake et al., 2023) could act contrary to user interests, such as by exfiltrating sensitive information. Standardized processes for undoing these actions could protect users.

- Minimizing contagion: Rollbacks could reduce the negative impacts of agents on third parties. For example, an agent could use inter-agent communication channels (Section 4.3) to spread text or image worms, which could hijack other agents and propagate themselves (Cohen et al., 2024; Gu et al., 2024). To help prevent further infection, administrators of these communication channels could unsend or take down the worm.

- Enabling productive interactions: Parties could make contracts or trade on the basis of rollbacks. For example, two users could allow their agents to engage in a financial transaction with the proviso that the transaction be reversed if an independent, objective third-party (such as an automatic monitoring system) deems one of the agents to have jailbroken the other.

**Existing alternatives**: Rollback infrastructure already exists in certain domains. For example, banks can void fraudulent transactions and social media companies can take down content. As agents come to carry out more tasks, potential challenges include making the infrastructure more accessible and responsive to users, scaling it to the potentially increased demand for rollbacks (e.g., if hijacking of agents becomes common), and building rollback infrastructure in domains where it does not already exist.

**Adoption**: Although rollbacks are likely to be useful, those responsible for undoing an action may be resistant to implementing them. This resistance may stem from the additional business uncertainty of a transaction that could be undone. For example, online merchants may have difficulty predicting demand if orders are undone at highly variable rates. If the advantages of rollbacks outweigh such disadvantages, governments could intervene to encourage their implementation. Analogously, US federal legislation allows consumers to dispute fraudulent credit card transactions (noa, 2023). Furthermore, monitoring and approval of rollback requests could be resource intensive in terms of human or AI labour.

**Limitations**: Some actions may be practically difficult to undo. For example, there is no way to undo physical harm caused by an agent operating a robotic system. Preventing such actions altogether or remedying them in another way (e.g., compensation) may be more appropriate.

Another potential issue is moral hazard. If rollbacks were readily available, developers could have weaker incentives to implement effective oversight layers. Similarly, users could have weaker incentives to properly monitor their agents. The cost of oversight failures would be transferred to those operating such rollback mechanisms. A potential way to deal with moral hazard is through insurance terms. For example, rollbacks could be made available only if users pay on insurance premium, deductibles, and/or co-payments.

**Research questions**:

- Could access to a rollback mechanism be made conditional upon some measure of the agent's reliability?

- How could rollbacks interact with insurance?

- How does implementation of rollbacks for agents affect business economics? Are the effects significantly different than those from existing rollback mechanisms (e.g., refund policies)?

- When is government intervention necessary to support implementation of rollbacks?

- How could rollbacks be abused?

# 6 Challenges and Limitations

We discuss challenges and limitations common to many types of agent infrastructure.

## 6.1 Adoption

A common challenge for all infrastructure is adoption. Some infrastructure mostly depends upon agent developers or deployers adopting it, such as oversight layers. Other infrastructure depends upon coordination between multiple parties. For example, two parties can only communicate if they use the same communication protocol. In these cases, network effects are likely to be crucial. Infrastructure is more likely to be widely adopted if the creators can take advantage of existing networks or build them. For example, Google has partnered with a number of large enterprises in the introduction of its recent agent-agent communication protocol (Surapaneni et al., 2025). Smaller and less well-known players could be less able to compete, even if their infrastructure offers superior features (also see Section 6.2). In addition to actors that build agent infrastructure, existing internet standards organizations could also be crucial for diffusion. Indeed, much agent infrastructure will build upon existing internet protocols, and some AI agent working groups already exist.[11]

## 6.2 Lack of Interoperability

Infrastructure from different developers could be mutually incompatible. For example, an ID system could fail to recognize IDs (or certifications) from competitor systems. Although this lack of interoperability would make IDs less useful, it could benefit dominant players by shutting out potential competitors. For example, an agent developer could try to promote use of its own agents by limiting the interoperability of its reputation system. Analogously, a restaurant cannot transfer its profile from Yelp to Google. Similar problems beset other digital platforms (e.g., Facebook Messenger, WhatsApp, iMessage). Legislation such as the EU's Digital Markets Act has encouraged more interoperability in these cases, but it remains to be seen whether and how similar rules could apply to the implementation of agent infrastructure.

## 6.3 Lock-In

Widespread adoption of insufficiently secure or otherwise harmful infrastructure could be difficult to reverse. Consider the network effects of interaction protocols. Moving away from a common protocol could mean losing important traffic. As an additional barrier, the benefits of moving away from the protocol could be realized only once a sufficient number of entities do so. Adoption of the Border Gateway Protocol (BGP) (Rekhter et al., 2006) ran into similar problems. Routing information on BGP is not verifiable: any entity can claim to have a route to another entity. In 2008, in an attempt to block YouTube domestically, a Pakistani ISP accidentally sent routing information that brought down YouTube worldwide (Mathew, 2014). Despite the potential for such problems, adoption of a verifiable version of BGP has been slow (Testart et al., 2024). For agents, it will be important to adopt procedures for effectively updating infrastructure.

# 7 Related Work

Past work has considered the protocols needed to create and manage networks of agents (Kahn & Cerf, 1988; Poslad, 2007; Wooldridge, 2009). While the specific details of such protocols may not be applicable to how today's agents operate, conceptual ideas such as a communication system between agents (Kahn & Cerf,

---

[11]E.g., see https://www.w3.org/groups/cg/webagents/

1988) are still relevant. Our paper builds upon this prior work by accounting for (1) unique features of more general-purpose, foundation-model-based agents and (2) modern digital infrastructure that could support agent infrastructure. Furthermore, we describe in detail how agent infrastructure can help to manage the risks of agents.

Some recent literature focuses on how to govern foundation-model-based agents. Chan et al. (2023); Lazar (2024); Gabriel et al. (2024); Kolt (2024) study the ethical and societal implications of agents. Shavit et al. (2023) propose practices for governing agents, some of which are related to types of agent infrastructure we discuss. Chan et al. (2024a;b) propose agent infrastructure that can help certain actors obtain visibility into how agents are used, whereas we take a broader view of the functions of agent infrastructure. Hammond et al. (2025) propose a taxonomy of risks from multi-agent systems. Finally, Perrier (2025); Perrier & Lazar (2025) study the ontology of agents and review historical examples of agent infrastructure.

We consider agent infrastructure to be a subset of **adaptation interventions** (Bernardi et al., 2024), which intervene on features other than the capabilities of an AI system. Our definition of agent infrastructure rules out adaptation interventions like laws, regulatory action, civic initiatives or activities, and applications of AI systems for a specific purpose (e.g., a company using an agent to find cybersecurity loopholes). At the same time, we expect agent infrastructure to help with the implementation of such adaptation interventions. For example, identity binding (Section 3.1) can aid the application of laws to AI agents.

## 8 Conclusion

We identified three functions for agent infrastructure: 1) tools for attributing actions, properties, and other information to specific agents, humans, or other actors; 2) methods for influencing agents' interactions with the world; and 3) systems for detecting and remedying harmful actions from agents. We proposed infrastructure that could help achieve each function and outlined use cases, adoption, limitations, and open questions. Our presentation is likely not exhaustive, and additional types of agent infrastructure may become relevant as agent capabilities improve.

Agent infrastructure is not itself a complete solution, but is rather a platform for policies and norms. For example, tools for identity binding by themselves do not resolve how to allocate liability for the actions of agents (Kolt, 2024; Wills, 2024). Our presentation of infrastructure has also neglected other challenges. For instance, if the deployment of agents causes significant unemployment (Korinek & Juelfs, 2022; Korinek & Suh, 2024), more comprehensive economic and political responses will be needed. Future work should study the regulatory and legal reforms needed to adapt to these and other challenges.

There are many opportunities for moving forward with agent infrastructure. Researchers could investigate open questions around the use and implementation of agent infrastructure. Agent developers, service providers, and other companies could begin to build and experiment with infrastructure. Building infrastructure that is interoperable, easy to use, and updatable will likely require engagement between governments, industry, and civil society. Analogous to the Internet Engineering Task Force for maintaining Internet infrastructure, specialized institutions could support agent infrastructure. As increasingly capable agents enter the world, agent infrastructure could enable their use in a way that promotes prosperity and safeguards society.

### Acknowledgments

For thoughtful feedback and discussions related to this work, we would like to thank: Sam Manning, Ben Bucknall, Peter Wills, Jide Alaga, Kevin Wang, Fazl Barez, Steven Adler, Helen Toner, Iason Gabriel, Zoë Hitzig, Sayash Kapoor, Sella Nevo, Dan Hendrycks, Usman Anwar, Ben Garfinkel, and Lewis Hammond.

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
