# OpenReview forum: "Infrastructure for AI Agents"
_TMLR — Accepted by TMLR_

### Review · Reviewer_chGj · 2025-03-05

**Summary Of Contributions:**

This paper presents the major elements of supporting infrastructure for
AI/LLM-based agents.  The unique issues presented by AI agents require elements
of infrastructure that are not currently present on the Internet or elsewhere.
For each of these unique issues, the authors present recommendations and
suggest future directions of study to continue the work on that particular
issue.

**Audience:**

Yes

**Broader Impact Concerns:**

I would like to see a paragraph or two discussing the issues that AI
infrastructure could present.  Some of these are touched on in Section 6, so it
could possibly be just a matter of expanding that section.  I think some issues
that were not mentioned would monopolies on agent operation, increased ease of
surveillance, negative impact on human welfare, marginalizing human use of the
Internet, etc. (some of these might be out of scope, so these are just
suggestions on what could be mentioned).

**Claims And Evidence:**

Yes

**Requested Changes:**

- [Sec. 2] I think there needs to be more discussion on scope here.  At first,
  I was thinking that this AgInfra would a framework for the whole Internet
  (akin to something like HTTP/S), but then one of the examples is talking
  about AgInfra as having an intra-company application.  Naturally, there are
  multiple ways in which this could be applied, but I think it is important to
  establish the primary application domains at the beginning: at least
  mentioned in the introduction and discussed in a bit more detail (paragraph
  or two) in Section 2.  This is probably discussed later, but the
  considerations for universal infrastructure vs intra-organizational
  infrastructure differ in some key ways.
  - [Sec 3.1] The question of scope comes up again for me here because I think
    the severity of the limitations on identity bindings are heavily dependent
    on the scope.  If we are talking about IB within a given service, it is
    definitely doable, but if we are looking for an Internet-wide or multi-site
    framework for IB, this is much harder implement (although maybe less so
    nowadays as many services require linking activity with an account (e.g.,
    how Google/Facebook acts like an SSO for many websites)).
  - [Sec. 6] Although this touched on briefly in Sec. 6, I think there should
    be a bit more discussion of the feasibility, overall, of the adoption of
    agent infrastructure.  All of these ideas are nice on paper, but in the
    realm of technology, we often find the pace of innovation outstripping the
    mechanisms for effectively organizing it.  I think one of the major reasons
    for this is that there are not incentives for the interventions even if
    they would be beneficial overall.  For that reason, it might be worth
    mentioning that a major meta-issue of agent infrastructure is tapping into
    incentives (maybe basing this on game theory); a robust discussion of this
    is probably beyond the scope of this paper, but it is a major issue that
    needs to be mentioned.  The feasibility of adoption is also dependent on
    the scope; e.g., adoption a within a company would be much more realistic
    than Internet-wide standards.
- [Sec 3.1] When we are linking an agent to an _action_, what is the action,
  itself.  Would it be something like a transaction on an e-commerce platform?
  It would seem like something as trivial as a mouse click wouldn't be
  recordable/traceable, so maybe it's not any action we care about but actions
  with a certain level of significance?
- [Sec. 3.2] Regarding adoption of auditing/certification: it would be
  worthwhile to mention government and other heavily-regulated domains where
  regulations often require certification of software products before they are
  used/deployed (e.g., [ISO
  9001:2015](https://www.iso.org/standard/62085.html))
- [Sec. 3.3] I would recommend mentioning URIs/URLs as they are an existing
  framework for identification and location of entities on networked systems,
  which is one aspect of agent identification that one would want to replicate.
- [Sec. 4.2, Existing alternatives] There should be more discussion here.  For
  example, online banking is a good example where activity in person is treated
  differently from online activity (analogous to the human/agent distinction,
  respectively).  Certain activity patterns trigger account freezes or manual
  review.
- [Sec. 4.2, Limitations] There should be more discussion here as well.  For
  example, compared to the other recommendations, oversight infrastructure
  seems like it could require far more investment of time and resources to
  work.  This is because oversight requires more of an active role of third
  parties compared to setting up an API or and ID system.
- [Sec. 4.3, Existing alternatives] This section could be a bit more thorough.
  What about something like RSS?  Of course, it is not tailored to agents but
  it seems like an existing format that already allows for easy dissemination
  of information among heterogeneous agents.
- [Sec. 4.3, Adoption] Expand.
- [Sec. 4.3, Limitations] "Scammers Could" -> "... could"
- [Sec. 4.3, Limitations] Again, this section is a bit sparse considering
  inter-agent communication is the place to discuss more extensively something
  that has been mentioned before: the possibility of agent viruses and worms.
  The Internet is what makes cybersecurity such a difficult and pressing topic
  and suggesting a way to smoothly interconnect agents through the Internet
  opens up a whole can of worms (haha) in terms of security and the potential
  for disaster.  This definitely needs to be expounded up on in this
  limitations section.
- [Sec. 4.3, Limitations] Might also be worth mentioning that inter-agent
  communication would greatly increase the resource utilization of agents since
  they no longer interacting with a user and some API but now with a whole host
  of other entities.  For example, imagine the difference in resource
  utilization between a web server (which receives many requests) and a word
  processor which basically only interacts with the user.
- [Sec. 4.4, Potential functions] Seems like it would be worth mentioning that
  prior elements of infrastructure could be enhanced by commitment devices such
  as oversight and certification.
- [Sec. 4.4, Existing alternatives] Expand on the mention of smart contracts as
  it seems like it could be the basis for the implementation of commitment
  devices for agents.
- [Sec. 4.4, Adoption] Expand.
- [Footnote 10] Training agents to "follow the law" is vastly underspecified as
  law itself underspecified and can only be made sense of the broader
  conditions of jurisprudence.
- [Sec. 5.1] Maybe mention
  [CVE](https://en.wikipedia.org/wiki/Common_Vulnerabilities_and_Exposures).
- [Sec. 5.2, Limitations] Expand.  One thing to possibly mention would be moral
  hazard of introducing rollbacks where actions are taken more carelessly
  because they can always be rolled back.
- [Sec. 5.2] In places where rolling back might be impractical, an alternative
  is "agent insurance" that can essentially insure responsible entities against
  mistakes made by its agent.  This would fit in with other elements of agent
  infrastructure like certification (e.g., reducing insurance risk), agent IDs,
  and attributability.  That being said, this isn't without major ethical
  implications as it introduces a moral hazard on par with that of rollbacks in
  the first place.  It seems worth mentioning, though.
- [Sec. 6.2, Difficult of Use] Expand.
- It seems like blockchain would a potential vehicle for some of the elements
  of infrastructure mentioned, but it was only alluded to via "smart
  contracts".

### Minor changes
- [p. 2] "blockers": this word is a bit unclear; either use a different
  word/phrase or give a brief definition by what is meant by it.
- [p. 2] "on their environments³ These": missing "."
- [Table 2] "AI agent safety": put a "\\" after "agent" to fix the awkward
  spacing
- [Sec 2.1] "etc" -> "etc."
- [Sec. 2.2] "Recourse": not clear what this word means; maybe use
  accountability, transparency, or attributability.
- [Sec 3.1] "AI agents are not legal entities.": In what jurisdiction(s)?
- [Sec. 3.2] Bold or italics typesets better than underlining, in my opinion.
- [Sec. 4.3, Research questions] "analogues" -> "analogous"

**Strengths And Weaknesses:**

### Strengths
- [major] I think the paper is thorough in addressing the relevant
  considerations of agent infrastructure.  For the most part, when developed
  a question or concern while reading the paper, it was quickly resolved as
  I continued to read.  The places where this was not successfully achieved are
  mentioned in the _Requested Changes_ section.
- [major] This paper is easy to follow and works well as a reference for the
  major issues in agent infrastructure.
- [minor] This paper is timely as it is clear that AI agents have significant
  potential while not having widespread use yet, potentially making some of the
  recommendations easier to implement.

### Weaknesses
- [major] Section 4 sort of dropped the ball compared to the rest of sections.
  I think this is mostly a matter of giving the thorough treatment to the
  topics as the other sections rather than needing to come up with radically
  new ideas.
- [minor] I think there are certain items that it would be natural to mention
  but were missed.  These are given below.

---

> ### Author Response · Authors · 2025-04-22
> **Appreciative of feedback, changes made**
>
> We thank the reviewer for their extremely detailed comments and useful feedback. We agree with the vast majority of the comments and have incorporated the feedback into the revised draft, as noted below (with a few clarification questions). In the draft, changes are in **blue**.
>
> # Changes
>
> > [Sec. 2] I think there needs to be more discussion on scope here. At first, I was thinking that this AgInfra would a framework for the whole Internet (akin to something like HTTP/S), but then one of the examples is talking about AgInfra as having an intra-company application. Naturally, there are multiple ways in which this could be applied, but I think it is important to establish the primary application domains at the beginning: at least mentioned in the introduction and discussed in a bit more detail (paragraph or two) in Section 2. This is probably discussed later, but the considerations for universal infrastructure vs intra-organizational infrastructure differ in some key ways.
>
> We have added a paragraph on scope at the end of section 2.1
>
> > [Sec 3.1] The question of scope comes up again for me here because I think the severity of the limitations on identity bindings are heavily dependent on the scope. If we are talking about IB within a given service, it is definitely doable, but if we are looking for an Internet-wide or multi-site framework for IB, this is much harder implement (although maybe less so nowadays as many services require linking activity with an account (e.g., how Google/Facebook acts like an SSO for many websites)).
>
> We have added additional text in 3.1 to this effect
>
> > [Sec. 6] Although this touched on briefly in Sec. 6, I think there should be a bit more discussion of the feasibility, overall, of the adoption of agent infrastructure. All of these ideas are nice on paper, but in the realm of technology, we often find the pace of innovation outstripping the mechanisms for effectively organizing it. I think one of the major reasons for this is that there are not incentives for the interventions even if they would be beneficial overall. For that reason, it might be worth mentioning that a major meta-issue of agent infrastructure is tapping into incentives (maybe basing this on game theory); a robust discussion of this is probably beyond the scope of this paper, but it is a major issue that needs to be mentioned. The feasibility of adoption is also dependent on the scope; e.g., adoption a within a company would be much more realistic than Internet-wide standards.
>
> We have added new subsection in section 6 (and removed/incorporated old subsection on ease of use that was only a sentence and comparatively shallow)
>
> > [Sec 3.1] When we are linking an agent to an action, what is the action, itself. Would it be something like a transaction on an e-commerce platform? It would seem like something as trivial as a mouse click wouldn't be recordable/traceable, so maybe it's not any action we care about but actions with a certain level of significance?
>
> We have added a research question in 3.1 about what the level of granularity might be
>
> > [Sec. 3.2] Regarding adoption of auditing/certification: it would be worthwhile to mention government and other heavily-regulated domains where regulations often require certification of software products before they are used/deployed (e.g., ISO 9001:2015)
>
> We have added a sentence to this effect in the adoption section of 3.2
>
> > [Sec. 3.3] I would recommend mentioning URIs/URLs as they are an existing framework for identification and location of entities on networked systems, which is one aspect of agent identification that one would want to replicate.
>
> We have added a sentence to this effect in the description section of 3.3
>
> Continued....

---

> > ### Author Response · Authors · 2025-04-22
> > **continued from the above**
> >
> > > [Sec. 4.2, Existing alternatives] There should be more discussion here. For example, online banking is a good example where activity in person is treated differently from online activity (analogous to the human/agent distinction, respectively). Certain activity patterns trigger account freezes or manual review.
> >
> > We have added an additional paragraph in the existing alternatives section of 4.2.
> >
> > > [Sec. 4.2, Limitations] There should be more discussion here as well. For example, compared to the other recommendations, oversight infrastructure seems like it could require far more investment of time and resources to work. This is because oversight requires more of an active role of third parties compared to setting up an API or and ID system.
> >
> > We have added additional discussion points about the limitations, in particular about human attention costs and the economic costs of using capable AI systems to do monitoring
> >
> > > [Sec. 4.3, Existing alternatives] This section could be a bit more thorough. What about something like RSS? Of course, it is not tailored to agents but it seems like an existing format that already allows for easy dissemination of information among heterogeneous agents.
> >
> > We have added more discussion of existing alternatives in the relevant section.
> >
> > > [Sec. 4.3, Adoption] Expand.
> >
> > We have added more discussion here and a reference to the expanded 6.1.
> >
> > > [Sec. 4.3, Limitations] "Scammers Could" -> "... could"
> >
> > Changed.
> >
> > > [Sec. 4.3, Limitations] Again, this section is a bit sparse considering inter-agent communication is the place to discuss more extensively something that has been mentioned before: the possibility of agent viruses and worms. The Internet is what makes cybersecurity such a difficult and pressing topic and suggesting a way to smoothly interconnect agents through the Internet opens up a whole can of worms (haha) in terms of security and the potential for disaster. This definitely needs to be expounded up on in this limitations section.
> >
> > We have added more discussion on the vulnerabilities of agents and communication channels that could otherwise work undesirably in section 4.3.
> >
> > > [Sec. 4.3, Limitations] Might also be worth mentioning that inter-agent communication would greatly increase the resource utilization of agents since they no longer interacting with a user and some API but now with a whole host of other entities. For example, imagine the difference in resource utilization between a web server (which receives many requests) and a word processor which basically only interacts with the user.
> >
> > This problem seems like it comes with agent usage in general? That is, increased user utilization is a direct outcome of demand for agent capabilities. Apologies if we have misunderstood.
> >
> > > [Sec. 4.4, Potential functions] Seems like it would be worth mentioning that prior elements of infrastructure could be enhanced by commitment devices such as oversight and certification.
> >
> > What might be an example?
> >
> > > [Sec. 4.4, Existing alternatives] Expand on the mention of smart contracts as it seems like it could be the basis for the implementation of commitment devices for agents.
> >
> > We have added more discussion in the relevant section.
> >
> > > [Sec. 4.4, Adoption] Expand.
> > We have added more discussion, particularly about the adoption of smart contracts.
> >
> > > [Footnote 10] Training agents to "follow the law" is vastly underspecified as law itself underspecified and can only be made sense of the broader conditions of jurisprudence.
> >
> > We are aware of a forthcoming work on this topic which we hope to cite, so we will delay addressing this comment.
> >
> > > [Sec. 5.1] Maybe mention CVE.
> >
> > Added reference in the description section
> >
> > > [Sec. 5.2, Limitations] Expand. One thing to possibly mention would be moral hazard of introducing rollbacks where actions are taken more carelessly because they can always be rolled back.
> >
> > We have added discussion on moral hazard and who bears the cost
> >
> > > [Sec. 5.2] In places where rolling back might be impractical, an alternative is "agent insurance" that can essentially insure responsible entities against mistakes made by its agent. This would fit in with other elements of agent infrastructure like certification (e.g., reducing insurance risk), agent IDs, and attributability. That being said, this isn't without major ethical implications as it introduces a moral hazard on par with that of rollbacks in the first place. It seems worth mentioning, though.
> >
> > We have added mention of insurance in the same section. Having to pay premiums is a way of dealing with moral hazard.
> >
> > > [Sec. 6.2, Difficult of Use] Expand.
> >
> > We have removed this section because it seems like an obvious enough problem, and have replaced it with a section on adoption in 6.1.
> >
> > > It seems like blockchain would a potential vehicle for some of the elements of infrastructure mentioned, but it was only alluded to via "smart contracts".
> >
> > We have added more discussion of blockchain in section 4.4.
> >
> >
> > continued..

---

> > > ### Author Response · Authors · 2025-04-22
> > > **Continued from the above**
> > >
> > > > [p. 2] "blockers": this word is a bit unclear; either use a different word/phrase or give a brief definition by what is meant by it.
> > >
> > > Switched to “barriers”
> > >
> > > > [p. 2] "on their environments³ These": missing "."
> > >
> > > Fixed
> > >
> > > > [Table 2] "AI agent safety": put a "\" after "agent" to fix the awkward spacing
> > >
> > > Fixed
> > >
> > > > [Sec 2.1] "etc" -> "etc."
> > >
> > > Fixed
> > >
> > > > [Sec. 2.2] "Recourse": not clear what this word means; maybe use accountability, transparency, or attributability.
> > >
> > > > Fixed
> > >
> > > > [Sec 3.1] "AI agents are not legal entities.": In what jurisdiction(s)?
> > >
> > > In any jurisdiction, to our knowledge.
> > >
> > > > [Sec. 3.2] Bold or italics typesets better than underlining, in my opinion.
> > >
> > > We mainly wanted to reserve bolding for definitions, but we will think about this feedback further.
> > >
> > > > [Sec. 4.3, Research questions] "analogues" -> "analogous"
> > >
> > > Here we do mean the noun “analogue” (https://www.merriam-webster.com/dictionary/analogue).

---

> > > > ### Author Response · Authors · 2025-04-22
> > > > **Continued from the above**
> > > >
> > > > > I would like to see a paragraph or two discussing the issues that AI infrastructure could present. Some of these are touched on in Section 6, so it could possibly be just a matter of expanding that section. I think some issues that were not mentioned would monopolies on agent operation, increased ease of surveillance, negative impact on human welfare, marginalizing human use of the Internet, etc. (some of these might be out of scope, so these are just suggestions on what could be mentioned).
> > > >
> > > > We have mentioned surveillance in the relevant sections (3.1, 3.3).
> > > >
> > > > We have added a line on monopolies on agent operation in 6.2.

---

### Review · Reviewer_aQmX · 2025-03-31

**Summary Of Contributions:**

This paper describes various infrastructure that could be built around AI agents to make their integration with other parts of society/the economy go more smooothly. They describe three main categories: identity verification, methods that allow people to intervene on potentially dangerous actions before they happen, and methods that allow potentially harmful actions to be detected after they've happened.

**Audience:**

Yes

**Claims And Evidence:**

Yes

**Requested Changes:**

I'm happy with the paper as is.

**Strengths And Weaknesses:**

I think the breakdown is fairly reasonable, and is fairly well-explained. (I appreciated the diagrams.)

A lot of the ideas here felt pretty obvious, but I guess that's to be expected when writing this kind of paper, and a neat categorization can be helpful even if all the basic ideas are simple.

In some places, I felt like the paper (like many governance papers) didn't sufficiently distinguish between questions that seem to have extremely easy, well-established answers (e.g. "how can agents verify themselves"--surely it suffices to use standard authentication software like OAuth for this?) and questions that require substantial future work (e.g. "For the example claims we list above, who should be making the claims? How could the claims be verified?").

I would like the paper better if it included concrete proposals of how you'd solve these problems by making use of current technology (e.g. how you'd use OAuth to do identity binding), and then described the extent to which the current technology leaves problems unsolved.

---

> ### Author Response · Authors · 2025-04-22
> **Changes made**
>
> We appreciate the positive and constructive feedback from the reviewer. Below, we outline additional changes we have made to address identified weaknesses. Changes within the draft are in **blue**.
>
> > In some places, I felt like the paper (like many governance papers) didn't sufficiently distinguish between questions that seem to have extremely easy, well-established answers (e.g. "how can agents verify themselves"--surely it suffices to use standard authentication software like OAuth for this?) and questions that require substantial future work (e.g. "For the example claims we list above, who should be making the claims? How could the claims be verified?").
>
> We have made a pass over the research questions to remove or modify those that seem overly trivial. We've highlighted such changes in **blue**.
>
> > I would like the paper better if it included concrete proposals of how you'd solve these problems by making use of current technology (e.g. how you'd use OAuth to do identity binding), and then described the extent to which the current technology leaves problems unsolved.
>
> We leave much of this for other/future work, but we have added more discussion of how existing technologies could help in sections 4.2, 4.3, and 4.4.

---

### Review · Reviewer_t2cz · 2025-04-16

**Summary Of Contributions:**

The paper introduces the concept of "agent infrastructure" as technical systems and shared protocols external to AI agents that mediate their interactions with environments. It contrasts this with system-level interventions (which modify the AI system itself) and argues that both approaches are necessary.
The authors categorize agent infrastructure into three functions: attribution (linking agents to identities and properties), interaction (shaping how agents engage with the world), and response (detecting and remedying harmful actions). For each category, they discuss potential implementations, adoption challenges, and open research questions.
The authors position agent infrastructure as necessary for managing AI agent ecosystems, drawing analogies to how Internet infrastructure (like HTTPS) and physical infrastructure (like traffic lights) enable safe and productive interactions.

**Audience:**

Yes

**Broader Impact Concerns:**

In addition to these changes, I would recommend to position the work more clearly within the broader landscape of AI governance and AI safety. Some specific questions:

1. The paper has limited discussion of the economic incentives for infrastructure adoption. What business models could support the development and maintenance of agent infrastructure? **Are there specific components where market incentives will be insufficient, requiring public investment or regulatory mandates?**
2. How might the proposed infrastructure evolve as AI capabilities advance? Which components would need to be redesigned for more capable systems, and which are likely to remain relevant? **What is the connection to long-term risks from increased automation?** Are specific design choices addressing aspects of existential risk or is building the infrastructure accelerating loss of human agency?

**Claims And Evidence:**

Yes

**Requested Changes:**

Overall it's a good paper. It provides a clear conceptual framework supported by logical arguments and references to existing systems, and it addresses an interesting topic.

1. I recommend to revisit the writing and delete the zero-information statements or tighten it to not risk loosing the reader mid-paper. (necessary)

There are several ways this paper could achieve more depth, it could be in a more clear priorization framework, deeper analysis of the architecture, or the governance questions or concrete technical frameworks.

2. More explicitly discuss how the proposed framework builds upon or differs from existing work in multi-agent systems, trustworthy AI, and internet infrastructure (recommended)

3. The catalogue-like structure of research questions after each section has more of a brainstorming approach. Consider consolidating these into a research roadmap with clear **priorities**: should include a framework for prioritizing which infrastructure components should be developed first based on necessity, feasibility, and potential impact. This will make the paper more actionable. (recommended)

4. Include more detailed examples of how specific infrastructure components might be implemented technically. (recommended)

5. The paper would benefit from more detailed discussion of how existing infrastructure in related domains (e.g., traditional software systems, internet protocols) has evolved to address similar challenges. (recommended)

6. As for the adoption, further explore the incentive structures that would lead different stakeholders to adopt the proposed infrastructure. (recommended)

**Strengths And Weaknesses:**

### Strengths

1. important topic, timely and interesting
2. well-written, and highly organized, balanced, essentially zero errors
3. concepts are clear, "agent infrastructure" as opposed to system-level interventions, three-function taxonomy (attribution, interaction, response)
	- authors situate their work relative to adaptation interventions (Bernardi et al., 2024)
	- examples in Table 1 illustrate each component of the framework, making abstract concepts more concrete
	- traffic safety analogy (comparing driver training to system-level interventions and traffic lights to infrastructure) is effective at making the distinction intuitive
4. good coverage, highly interdisciplinary
	- spans a wide range of infrastructure types and considerations across multiple domains
	- identity binding section addresses both accountability mechanisms and privacy considerations, drawing parallels to existing identity verification systems like those in ridesharing.
	- attention to detail when discussing potential implementations
	- coverage of agent channels demonstrates awareness of both existing software interfaces and dedicated internet infrastructure possibilities.
5. practical focus
	- focus on implementation rather than just theoretical concepts
		- e.g. certification section (3.2) discusses specific verification mechanisms and potential claims that could be included in certificates, such as "the tools the agent can access" and "how an agent handles sensitive information."
	- assesses adoption challenges
6. foresight
	- identify challenges that will become increasingly important as agent capabilities advance.
		- e.g. inter-agent communication (4.3) or commitment devices section (4.4) shows foresight about potential future collective action problems
	- rollbacks (5.2), which anticipates the need for mechanisms to undo agent actions in cases of malfunction or hijacking

###  Weaknesses

The central weakness of this paper is the lack of depth.

The paper frequently relies on statements that sound reasonable but lack substance or insight. For example, statements like "agent infrastructure will be similarly indispensable to ecosystems of agents" (p.3) and "tools that address these blockers could enable positive uses of agents" (p.2) or "we will need tools both to unlock their benefits and manage their risks" are conceptually true but offer minimal analytical depth. The paper has too many of such essentially zero-information statements.

The following are directions that could be further explored for more insight.
1. lack of empirical foundation
	- while the paper acknowledges that much of the proposed infrastructure is speculative, it would be significantly strengthened by more empirical grounding.
	- some claims about the necessity of certain infrastructure components could be better supported with more concrete examples or case studies, e.g. from adjacent fields.
		- e.g. discussion of identity binding could examine the specific technical mechanisms of existing digital identity solutions like Estonia's e-Residency program or India's Aadhaar system, analyzing their strengths and limitations for agent applications
		- similarly, the certification section could analyze specific technical implementations of existing certification systems like SSL/TLS in more detail
2. incomplete analysis of implementation
	- While implementation challenges are mentioned throughout, they often lack sufficient technical depth.
		- e.g. discussing oversight layers (4.2), the paper mentions "security or confidentiality" considerations but doesn't explore the specific technical challenges of implementing secure oversight interfaces that can't be circumvented by sophisticated agents.
		- discussion of inter-agent communication (4.3) acknowledges potential for abuse but doesn't analyze fundamental technical challenges like ensuring message authenticity or preventing DoS attacks in broadcast functionalities.
	- more detailed analysis would consider specific architectural approaches (centralized vs. decentralized, push vs. pull) and their respective security implications.
3. unclear prioritization
	- presents numerous infrastructure components without providing a framework for prioritizing development. For example, the paper doesn't clearly identify whether identity binding (3.1) or agent channels (4.1) should be developed first or by whom, or which combination of infrastructure components would provide a minimal viable ecosystem for safe agent deployment. What stakeholders may develop it, and which components researcher should focus on to work for the public interest (certain components may be developed anyways by private actors for example)
	- also, the current structure doesn't distinguish between infrastructure components that would be immediately valuable for current LLM agents versus those that become important only as capabilities advance. For example, commitment devices (4.4) may be more relevant for future advanced agents than for current systems, while incident reporting (5.1) could be needed today.
4. insufficient integration with existing technical standards
	- mentions that some infrastructure could build upon existing systems (like using OpenID for identity binding), it doesn't sufficiently analyze how existing technical standards and protocols could be extended or modified. For instance, when discussing agent IDs (3.3), the paper could analyze how W3C's Decentralized Identifiers (DIDs) or Verifiable Credentials standards might be adapted, or how the OAuth framework could be extended for agent-specific authorization flows.
	- inter-agent communication section (4.3) could benefit from a discussion of how existing communication protocols like ActivityPub (used by Mastodon) or Matrix might be adapted for agent use cases, rather than implying entirely new protocols would need to be developed.
5. underexplored governance dimensions
	- the paper mentions that agent infrastructure is not itself a complete solution but rather a platform for policies and norms, it doesn't adequately develop this connection. For example, the certification section (3.2) mentions the potential for adverse selection (as seen with TrustARC), but doesn't sufficiently explore governance mechanisms that could prevent similar failures in agent certification.
	- who should develop and maintain these infrastructure components? Should they be developed by individual companies, industry consortia, standards bodies, or government agencies? What governance structures would ensure these infrastructure components serve the public interest while remaining technically sound? What are the components that are most interesting for maintaining human agency in the age of increased automation?

### Minor
- clashing color scheme with red and light orange, consider two colors that are more harmonious (if going for a binary color scheme)
- "fradulently" should be "fraudulently"
- some references include page numbers while others don't, in general a bit inconsistent references, with author names and sometimes URLs sometimes not
- "Could use an addressing system" incorrectly capitalizes "Could" mid-paragraph
- "Analogously, hospitals separate networks for storing" is missing "some" or similar word
- figure 3: arrows between components don't clearly indicate the direction of information flow

---

> ### Author Response · Authors · 2025-04-22
> **Changes made**
>
> We thank the reviewer for their detailed and constructive feedback on our paper. Such feedback has also inspired new research directions for us. We outline below the changes that we have made to our draft. In the revised draft, changed text is in **blue**.
>
> # changes
>
> > I recommend to revisit the writing and delete the zero-information statements or tighten it to not risk loosing the reader mid-paper. (necessary)
>
> We have addressed the specific examples referenced in the review and have also made a pass throughout the paper for removing zero-information statements, in particular in the abstract and introduction.
>
> > More explicitly discuss how the proposed framework builds upon or differs from existing work in multi-agent systems, trustworthy AI, and internet infrastructure (recommended)
>
> We have added an additional paragraph in section 7 on this point
>
> > The catalogue-like structure of research questions after each section has more of a brainstorming approach. Consider consolidating these into a research roadmap with clear priorities: should include a framework for prioritizing which infrastructure components should be developed first based on necessity, feasibility, and potential impact. This will make the paper more actionable. (recommended)
>
> We have added a new section 2.3 that addresses which agent infrastructure to prioritize.
>
> > Include more detailed examples of how specific infrastructure components might be implemented technically. (recommended)
>
> While we leave much of this detail for other/future work, we have added more discussion of how specific infrastructure components could integrate with existing technologies in sections 4.2, 4.3, and 4.4.
>
> > The paper would benefit from more detailed discussion of how existing infrastructure in related domains (e.g., traditional software systems, internet protocols) has evolved to address similar challenges. (recommended)
>
> See above.
>
> > As for the adoption, further explore the incentive structures that would lead different stakeholders to adopt the proposed infrastructure. (recommended)
>
> We have added more discussion of this point both within individual sections and in the new section 6.1.
>
> > The paper has limited discussion of the economic incentives for infrastructure adoption. What business models could support the development and maintenance of agent infrastructure? Are there specific components where market incentives will be insufficient, requiring public investment or regulatory mandates?
>
> While we consider much of these questions to be future (and upcoming) work, we have added the new section 6.1 to address questions of adoption.
>
> > clashing color scheme with red and light orange, consider two colors that are more harmonious (if going for a binary color scheme)
>
> Noted. We have removed the light orange colours.
>
> > "fradulently" should be "fraudulently"
>
> Fixed
>
> > some references include page numbers while others don't, in general a bit inconsistent references, with author names and sometimes URLs sometimes not
>
> Fixed
>
> > "Could use an addressing system" incorrectly capitalizes "Could" mid-paragraph
>
> Fixed
>
> > "Analogously, hospitals separate networks for storing" is missing "some" or similar word
>
> Fixed
>
> > figure 3: arrows between components don't clearly indicate the direction of information flow
>
> Fixed

---

> > ### Comment · Reviewer_t2cz · 2025-04-22
> > **looks good! one more small fix**
> >
> > Just noted another minor error that's introduced in blue: "to Through insurance terms" → lower case "through". Looks good otherwise.

---

> > > ### Author Response · Authors · 2025-04-22
> > >
> > > thanks! fixed now

---

### Decision · Action_Editor_8Ay2 · 2025-05-09

**Recommendation:** Accept as is

**Comment:**

The latest version of the paper appears to be in good shape and all issues identified by reviewers appear to be addressed so no revisions are needed other than the removal of the color used to track changes for reviewers and the standard changes to produce the camera ready version.

**Audience:**

All three reviewers agree this standard is met and I agree.  The framework may provide value as a way to think about challenges and solutions for those interesting in building or using agents.

**Claims And Evidence:**

Reviewers noted a number of places the paper could be improved.  After the authors updated the paper and responded all three reviewers were satisfied that the updated paper met this standard.  Overall the paper provides a clear argument for its concept of agent infrastructure and supports it with appropriate examples and references to the literature.